# Ultra-broadband optical amplification using nonlinear integrated waveguides

Ping Zhao[1,2✉], Vijay Shekhawat[1], Marcello Girardi[1], Zonglong He[1], Victor Torres-Company[1] & Peter A. Andrekson[1✉]

Four-wave mixing is a nonlinear optical phenomenon that can be used for wideband low-noise optical amplification and wavelength conversion. It has been extensively investigated for applications in communications[1], computing[2], metrology[3], imaging[4] and quantum optics[5]. With its advantages of small footprint, large nonlinearity and dispersion-engineering capability, optical integrated waveguides are excellent candidates for realizing high-gain and large-bandwidth four-wave mixing for which anomalous dispersion is a key condition. Various waveguides based on, for example, silicon, aluminium gallium arsenide and nonlinear glass have been studied[6–10], but suffer from considerable gain and bandwidth reductions, as conventional design approaches for anomalous dispersion result in multi-mode operation. We present a methodology for fabricating nonlinear waveguides with simultaneous single-mode operation and anomalous dispersion for ultra-broadband operation and high-efficiency four-wave mixing. Although we implemented this in silicon nitride waveguides, the design approach can be used with other platforms as well. By using higher-order dispersion, we achieved unprecedented amplification bandwidths of more than 300 nm in these ultra-low-loss integrated waveguides. Penalty-free all-optical wavelength conversion of 100 Gbit s$^{-1}$ data in a single optical channel of over 200 nm was realized. These single-mode dispersion-engineered nonlinear waveguides could become practical building blocks in various nonlinear photonics applications.

With the distinct advantages of overcoming the bandwidth, noise figure and wavelength range of a stimulated-emission optical amplifier and generating waves beyond those achievable with conventional lasers, four-wave mixing (FWM) has led to numerous applications in various fields. In particular, hyper-dispersion engineering (second- and fourth-order dispersion in tandem) is very critical for broadband FWM with parametric gain, which is being pursued in various areas, such as ultra-long-haul transmission[11], all-optical high-speed signal processing[12], light detection and ranging[13], and biochemistry analysis[14]. Since the invention of low-loss silica fibres, which offer long nonlinear optical interaction distances, fibre-based FWM has been intensively investigated[15]. However, the fibres exhibit low nonlinearity as well as a narrow parametric bandwidth and suffer from polarization and dispersion drifts, which reduce both FWM efficiency and bandwidth[16]. Because of their high nonlinearity, small footprint and flexible patterning, $\chi^{(3)}$-based nonlinear integrated waveguides provide excellent on-chip control of the optical field properties, such as dispersion and polarization state, potentially paving the way to high-efficiency and wideband FWM[17]. Nonlinear semiconductor integrated waveguides with a low refractive index contrast formerly attracted much interest for use in FWM but suffer from limited dispersion engineering and severe interference from other nonlinear effects[18]. Advances in deposition and bonding have enabled the creation of high-refractive-index-contrast nonlinear integrated waveguide structures with silica cladding that offer strong field confinement and dispersion engineering[19] and are ideally suited for broadband parametric signal processing. Many kinds of such nonlinear platforms have been explored, including silicon[6,20,21], silicon nitride[22–24], aluminium gallium arsenide[7,25,26], nonlinear glasses[9,10] and graphene[27]. In particular, a continuous-wave optical parametric gain due to FWM was achieved for the first time recently in nonlinear $Si_3N_4$ integrated waveguides, due to a balance of propagation loss, nonlinearity, power handling ability and dispersion engineering[22], which also corresponds to high conversion efficiencies (the power ratio of the output idler to input signal).

For pump-degenerate FWM, a strong pump wave (p) and a weak signal wave (s) were passed into a $\chi^{(3)}$-based nonlinear optical medium, where the signal was amplified and an idler wave was generated at an angular frequency of $\omega_i = 2\omega_p - \omega_s$, as illustrated in Fig. 1c. The phase mismatch parameter $\Delta K = \sum_{k=1}^{\infty} 2\beta_{2k}\Delta\omega^{2k}/(2k)! + 2\gamma P$ affects the gain, conversion efficiency and bandwidth. Here $\beta_i$ is the $i$th-order derivative with respect to the angular frequency $\omega$ of the optical propagation constant $\beta$ evaluated at the pump frequency, $\gamma$ is the nonlinear coefficient, $\Delta\omega$ is the angular frequency difference between the pump and signal waves and $P$ is the pump power[28]. Anomalous dispersion ($\beta_2 < 0$) is of vital importance for realizing phase matching ($\Delta K = 0$) for high parametric gain and wide bandwidth, as can be seen in the Supplementary Information. However, conventional high-index-contrast silica-clad nonlinear integrated waveguides are multi-mode to achieve

[1]Photonics Laboratory, Department of Microtechnology and Nanoscience, Chalmers University of Technology, Gothenburg, Sweden. [2]College of Electronics and Information Engineering, Sichuan University, Chengdu, China. ✉e-mail: zhao.ping@scu.edu.cn; peter.andrekson@chalmers.se

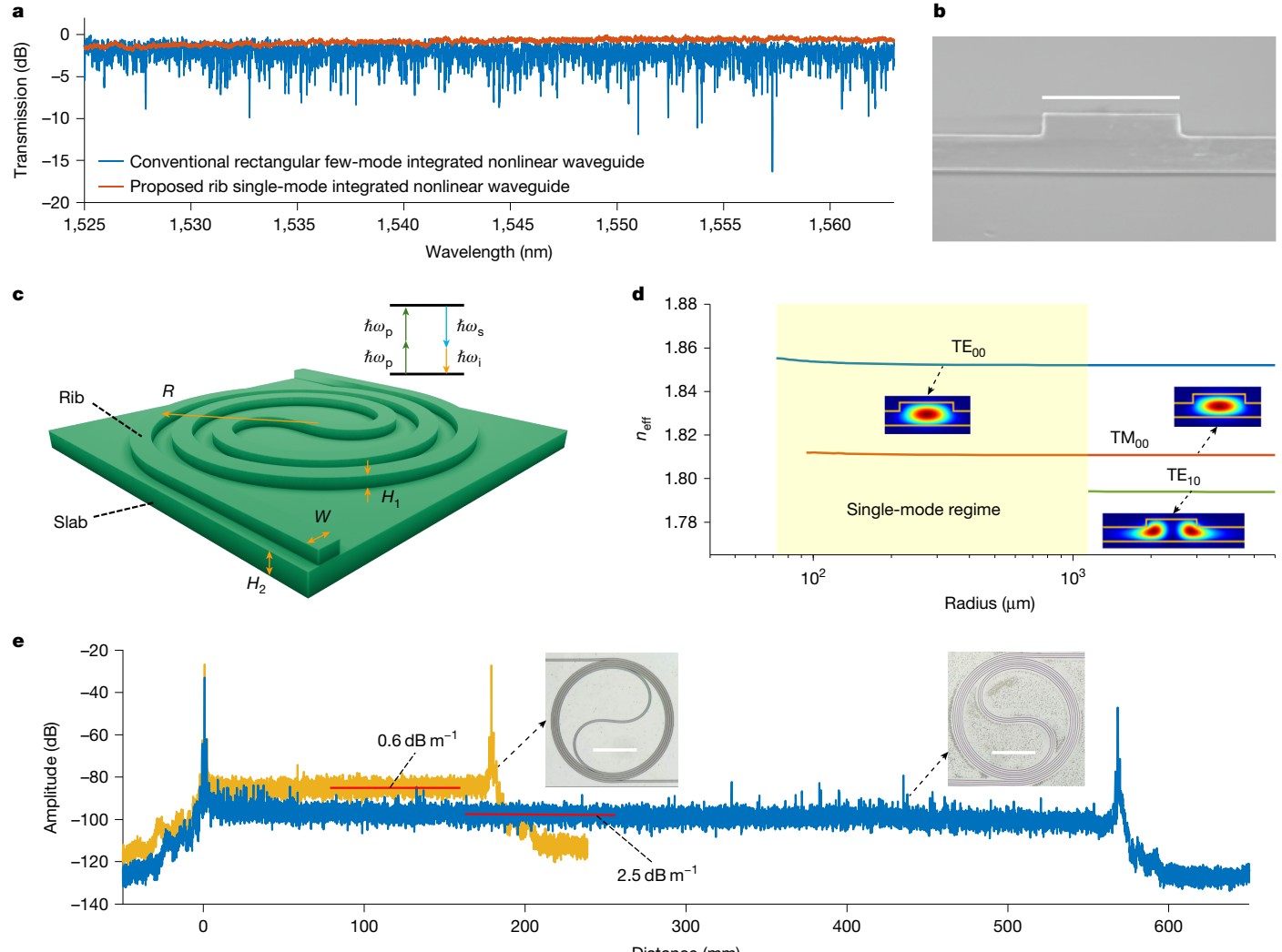

**Fig. 1 | Single-mode dispersion-engineered nonlinear integrated waveguides for ultra-broadband optical amplification and wavelength conversion.** **a**, Normalized measured transmission spectra of a conventional rectangular-core (blue) and a proposed single-mode rib (red) nonlinear $Si_3N_4$ integrated waveguide. The fabrication of the few-mode rectangular-core nonlinear spiral $Si_3N_4$ integrated waveguide was optimized to reduce sidewall roughness. The wavelength tuning step in the measurements was 1 pm. Both waveguides were about 50 cm long. **b**, Scanning electron microscope image of the cross section of one proposed single-mode nonlinear rib $Si_3N_4$ integrated waveguide. **c**, Schematic diagram of one unit of the proposed spiral single-mode dispersion-engineered rib nonlinear integrated waveguide. The maximum bend radius in the spiral area was controlled to simultaneously achieve single-mode operation and anomalous dispersion. Several spiral units were concatenated to generate a metres-long nonlinear integrated waveguide. The cladding was $SiO_2$. **d**, Effective refractive index ($n_{eff}$) of different modes in a nonlinear rib $Si_3N_4$ integrated waveguide versus the bend radius, with $W = 1.9$ μm, $H_1 = 300$ nm and $H_2 = 500$ nm. The blue, red and green lines are for the $TE_{00}$, $TM_{00}$ and $TE_{10}$ modes, respectively. Insets, intensity profiles of the different modes. The yellow lines are the $SiO_2$–$Si_3N_4$ boundaries. **e**, OFDR traces of 18-cm-long (yellow, WG1) and 56-cm-long (blue, WG2) single-mode nonlinear rib $Si_3N_4$ integrated waveguides when the wavelength was scanned from 1,480 nm to 1,640 nm. The minimal propagation losses of WG1 and WG2 were 0.6 dB m$^{-1}$ and 2.5 dB m$^{-1}$ (red lines). Insets, optical microscope images of one spiral unit of WG1 and WG2 on two wafers, respectively. Scale bars, 1.9 μm (**b**), 0.3 mm (**e**).

anomalous dispersion[6,7,10,22,23], but the focus has only been on the transverse cross-sectional geometry of the waveguide. Consequently, random and unavoidable modal coupling results in power drops for both the signal and pump waves, which not only decreases the FWM gain, conversion efficiency and bandwidth but also distorts the modulated signals[29]. For instance, the blue curve in Fig. 1a is the measured normalized transmission spectrum of a conventional rectangular-core dispersion-engineered nonlinear $Si_3N_4$ integrated waveguide whose fabrication was optimized to reduce the sidewall roughness. This waveguide was 2,000 nm wide, 690 nm high and about 50 cm long and supported four modes in transverse electric polarization. As can be seen in Fig. 1a, this typical conventional nonlinear $Si_3N_4$ integrated waveguide suffers from serious spectral fluctuations (power fading at some wavelengths of more than 10 dB) due to random mode coupling.

Single-mode $\chi^{(3)}$-based high-index-contrast silica-clad nonlinear integrated waveguides with simultaneous anomalous dispersion are in great need for FWM but have not been reported so far. Moreover, hyper-dispersion engineering ($\beta_4$ in tandem with $\beta_2$) is quite important for ultimately broadening the FWM bandwidth. Nevertheless, ultra-broadband FWM with high conversion efficiency assisted by hyper-dispersion engineering in single-mode nonlinear waveguides has not yet been demonstrated.

We propose a universal design method for achieving anomalous-dispersion single-mode nonlinear integrated waveguides to address the above issues. The method combines longitudinal bending with a transverse cross-sectional construction. The red curve in Fig. 1a is the measured transmission spectrum of one waveguide that we designed and fabricated (Methods). It has an excellent single-mode

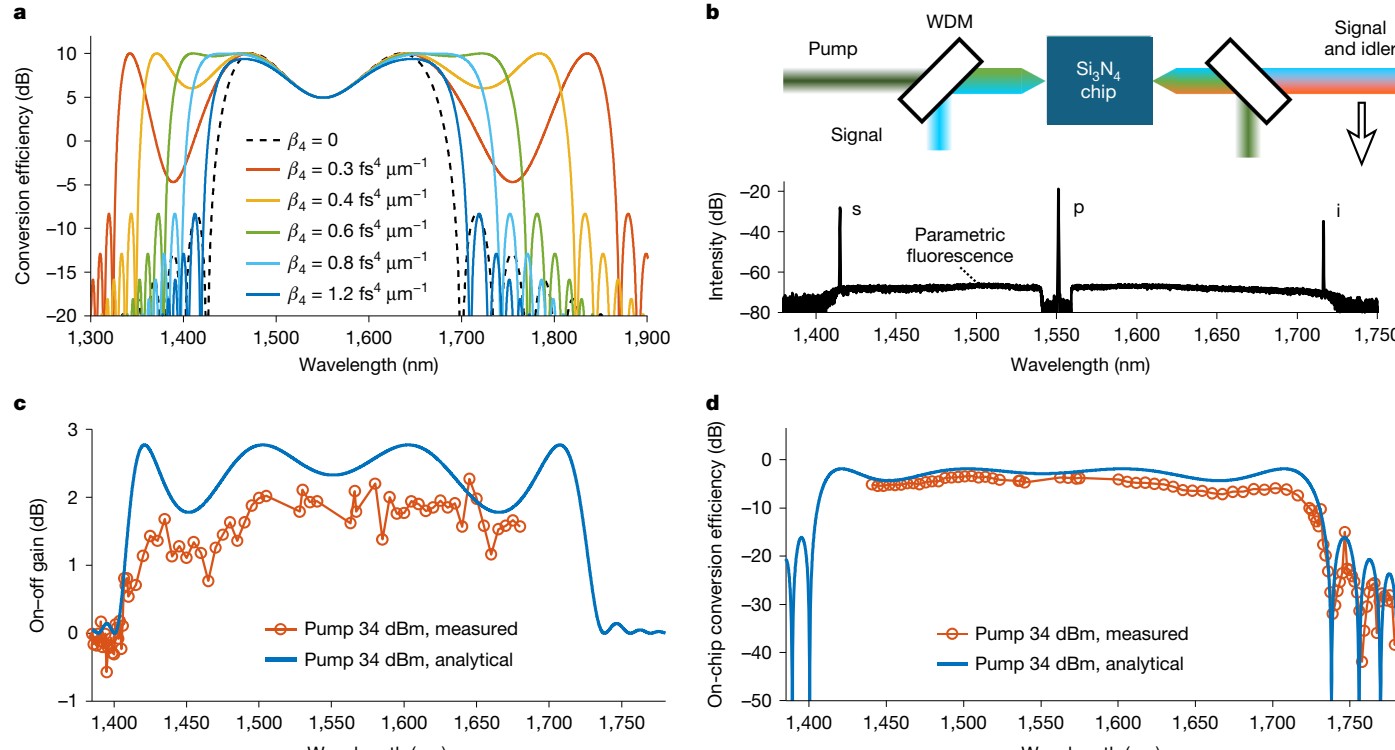

**Fig. 2 | Ultra-broadband integrated parametric waveguides with hyper-dispersion engineering. a**, Theoretical conversion-efficiency spectral curves of 1-m-long $\chi^{(3)}$-base nonlinear integrated waveguides with various fourth-order dispersions. The waveguide loss was 1 dB m$^{-1}$, the nonlinear coefficient was 0.7 W$^{-1}$ m$^{-1}$ and the second-order dispersion was $-1$ ps$^2$ km$^{-1}$. The power at 1,550 nm wavelength was 35 dBm. The red, yellow, green, light blue and dark blue solid lines are for $\beta_4 = 0.3$, 0.4, 0.6, 0.8 and 1.2 fs$^4$ μm$^{-1}$, respectively. The black dashed line corresponds to the case where fourth-order dispersion is not considered. **b**, Top, experimental set-up for the FWM characterization of the 0.56-m-long single-mode nonlinear rib Si$_3$N$_4$ integrated waveguide. Bottom, spectrum after the WDM coupler with a 1,551.1 nm pump and a 1,415 nm signal. **c**,**d**, Measured (red circles) on–off parametric gain (**c**) and on-chip conversion efficiency (**d**) of the single-mode nonlinear Si$_3$N$_4$ integrated waveguide with a 34-dBm on-chip pump power. The solid blue curves were calculated analytically with the assumption of a spectrally constant waveguide loss of 2.5 dB m$^{-1}$.

property in contrast to the conventional rectangular-core nonlinear integrated waveguide. Rib waveguides with silica cladding are used to achieve fewer guiding modes and lower propagation losses compared to rectangular-core waveguides with the same width and total thickness[19,30,31].

Figure 1b is a scanning electron microscope image of the cross section of a proposed 1.9-μm-wide single-mode nonlinear rib Si$_3$N$_4$ integrated waveguide. The key technique for simultaneously achieving single-mode operation and anomalous dispersion is to bend the waveguide to cut off higher-order modes and maintain the anomalous dispersion. Figure 1c is a schematic diagram of the proposed waveguide. $W$, $R$, $H_1$ and $H_2$ are the rib width, radius, height and slab thickness, respectively. To verify the proposed method, we used the Si$_3$N$_4$ integrated platform as an example to realize single-mode dispersion-engineered nonlinear waveguides for parametric gain. Figure 1d presents the simulated effective refractive index ($n_{\mathrm{eff}}$) of different modes as a function of waveguide radius at wavelength 1,550 nm. In the simulation, the nonlinear rib Si$_3$N$_4$ waveguide was 1.9 μm wide with $H_1 = 300$ nm and $H_2 = 500$ nm. The blue and red lines correspond to the fundamental transverse-electric and transverse-magnetic modes. As can be seen in Fig. 1d, when it is straight, this nonlinear rib Si$_3$N$_4$ waveguide supports three modes. When the radius was reduced to less than 1,150 μm, the high-order mode TE$_{10}$ (green line) was cut off. Hence, with the right bend arrangement, we obtained a nonlinear spiral-rib Si$_3$N$_4$ waveguide with a single mode per polarization. Extended Data Fig. 1 presents the simulated effective refractive index versus wavelength. The cutoff wavelength of the TE$_{10}$ mode was 1,180 nm for a bending radius of 400 μm. We show in the Supplementary Information how the propagation of the

TE$_{00}$ and TE$_{10}$ modes was affected by bending. With a larger nonlinear coefficient than the TM$_{00}$ mode, we used the TE$_{00}$ mode for FWM. The TE$_{00}$ mode dispersion at 1,550 nm could become anomalous and tuned by changing the waveguide width and radius (Extended Data Fig. 2a,b).

For the fabrication tolerance analysis, Extended Data Fig. 2c,d presents the second- and fourth-order dispersion for small rib dimensions. Moreover, Fig. 1e shows the traces of optical frequency-domain reflectometry (OFDR) of two TE$_{00}$-mode-coupled 1.9-μm-wide nonlinear spiral-rib Si$_3$N$_4$ integrated waveguides with lengths of 18 cm (yellow, WG1) and 56 cm (blue, WG2). The slab widths of WG1 and WG2 were sufficiently large such that the TE$_{00}$ mode was not affected by the slab sidewall and did not couple to adjacent rib waveguides (Supplementary Information). The red line in Fig. 1e indicates that the measured propagation loss of WG1 was 0.6 dB m$^{-1}$. For WG2, the propagation loss was about 2.5 dB m$^{-1}$. The insets in Fig. 1e are optical microscope images of WG1 and WG2, respectively. One can see that WG2 suffered from residual nanoparticles of which WG1 was almost free. The propagation loss difference between WG1 and WG2 was mainly due to fabrication variation, which we are trying to improve. We fabricated six 56-cm-long single-mode nonlinear rib Si$_3$N$_4$ integrated waveguides. WG2 was the only one without large defects on the OFDR traces. The yield for the 18-cm-long rib waveguides was 4/20, which was mainly limited by the misalignment of the dual-layer tapers and minor defects.

Apart from the single-mode waveguiding property, we investigated how the hyper-dispersion engineering ultimately extended the bandwidth of the FWM-based parametric process. Figure 2a depicts the theoretical conversion-efficiency spectra of a 1-m-long nonlinear integrated waveguide with a pump power of 35 dBm at 1,550 nm. In the

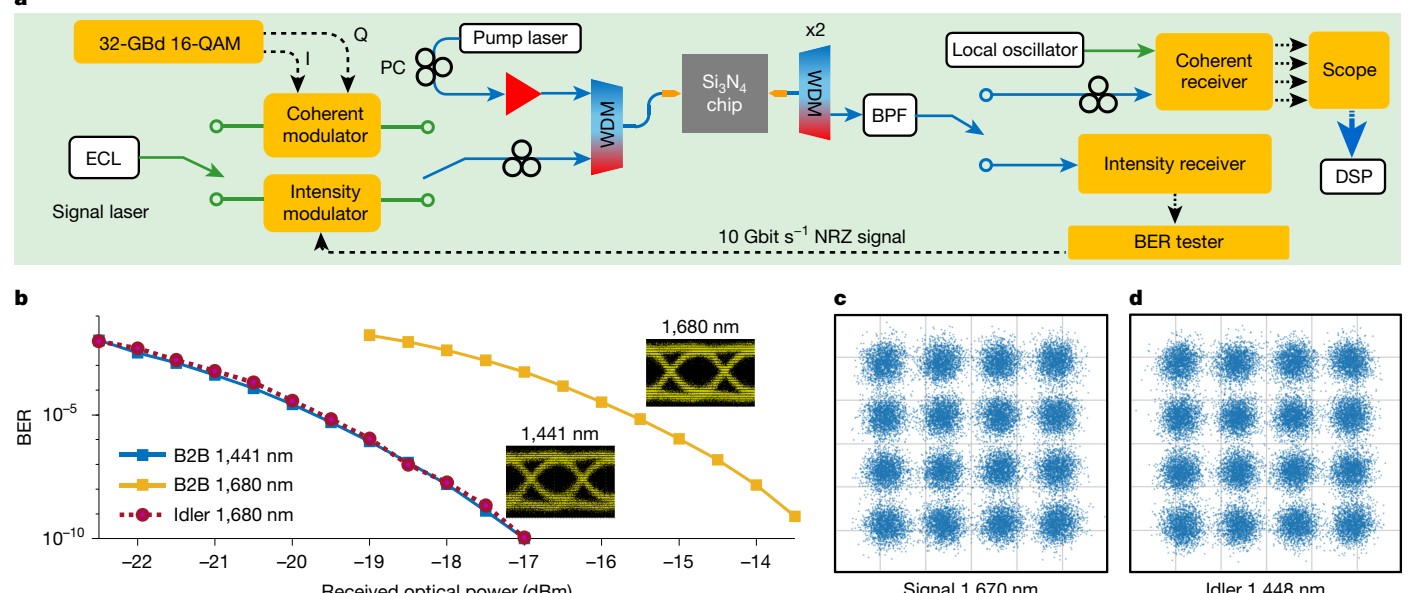

**Fig. 3 | Ultra-broadband high-efficiency, high-speed all-optical wavelength conversion based on single-mode nonlinear spiral-rib Si₃N₄ integrated waveguides. a**, Experimental set-up of the Si₃N₄-chip-based all-optical wavelength conversion for both intensity modulation and coherent optical communications. The on-chip pump power was 34 dBm. **b**, BER as a function of received optical power of 10 Gbit s⁻¹ NRZ signals (1,680 nm) and converted idlers (1,441 nm, purple dashed line). Insets, eye diagrams of the B2B optical signals at wavelengths of 1,441 nm and 1,680 nm, respectively. **c**,**d**, Constellation diagrams for a 1,670-nm signal (**c**) and a 1,448-nm idler (**d**) with 32-GBd 16-QAM after offline digital signal processing (DSP). BPF, band-pass filter; PC, polarization controller.

calculation, the second-order dispersion was −1 ps² km⁻¹ at 1,550 nm, the effective nonlinear coefficient was 0.7 W⁻¹ m⁻¹ and the waveguide loss was 1 dB m⁻¹. The maximum conversion efficiency was 10 dB, which corresponds to a maximum parametric gain of about 10 dB for the signal wave[32]. The parametric gain spectrum was like the spectral curve for conversion efficiency in this case. The black dashed line is for $\beta_4 = 0$ for which the amplification bandwidth was 270 nm. The red, yellow, green and blue solid lines are for $\beta_4 = 0.3$, 0.4, 0.6 and 1.2 fs⁴ μm⁻¹, respectively. As shown by Fig. 2a, the amplification bandwidth increased to 542 nm for $\beta_4 = 0.3$ fs⁴ μm⁻¹, as the fourth-order dispersion led to new phase-matching wavelengths[28]. When the balance among the nonlinear shift, second- and fourth-order dispersion was achieved with $\beta_4 = 0.6$ fs⁴ μm⁻¹, two flat gain regimes were obtained. The amplification bandwidth reached 385 nm (43% bandwidth increase compared to $\beta_4 = 0$). Hence, fourth-order dispersion plays a vital role in realizing ultra-wideband parametric devices.

Furthermore, we characterized the ultra-broadband FWM in WG2 using continuous-wave pump–probe approaches based on the experimental diagram in Fig. 2b (Methods). The on-chip pump power was 34 dBm considering the coupling loss. The lower part of Fig. 2b shows the output optical spectrum of WG2 with the residual pump mitigated by a wavelength-division multiplexing (WDM) coupler. The signal, pump and idler wavelengths were 1,415, 1,551.1 and 1,716 nm, respectively. Ultra-wideband flat parametric fluorescence during FWM was also observed, as can been seen in Fig. 2b. Figure 2c,d depicts the measured (blue) on–off parametric gain and on-chip conversion-efficiency spectra, respectively. The on–off gain was used as it can mitigate the impact of the wavelength-dependent coupling loss of the tapers on the measurements. The solid lines are theoretically fitted spectra with $\beta_2 = -2.2$ ps² km⁻¹ and $\beta_4 = 1.9$ fs⁴ μm⁻¹ at 1,551 nm. The measured and theoretical curves are in good agreement with small discrepancies, which may be due to the wavelength-dependent loss of the waveguide. The on-chip waveguide loss in the L band was about 1 dB, indicating that we achieved an on-chip net continuous-wave parametric gain of 1 dB. Besides, we obtained a maximum on-chip conversion efficiency

of −3.4 dB at 1,500-nm wavelength, as shown in Fig. 2d. The gain and conversion-efficiency spectra in Fig. 2c,d indicate that we realized a FWM bandwidth of 330 nm—one of the widest bandwidths of all reported continuous-wave optical amplifiers to date. As there were not enough lasers to cover the full FWM bandwidth during the measurements, we recorded the pure parametric-fluorescence spectrum as a measure of the parametric gain profile, which changed with the dispersion by adjusting the pump wavelength (Supplementary Information). Moreover, the fitted second- and fourth-order dispersion agreed with the waveguide design, as we can see from Extended Data Fig. 2c,d, which verifies that the nonlinear Si₃N₄ integrated platform is very tolerant to fabrication uncertainties. $\beta_2$ was more sensitive to variations in the thickness, whereas $\beta_4$ exhibited a larger tolerance to dimension variations, as can be seen from Extended Data Fig. 2c,d. The fitted $\beta_2$ is slightly smaller than the designed value of −4 ps² km⁻¹, which is mainly attributed to the thickness variation of the waveguide. Si₃N₄ wafers will be planarized to improve the thickness uniformity[33].

Furthermore, we applied the Si₃N₄-chip ultra-broadband efficient FWM to all-optical high-speed wavelength conversion for communications. Figure 3a shows the experimental set-up with intensity and coherent modulation (Methods). We used 10 Gbit s⁻¹ non-return-to-zero (NRZ) intensity modulation to check the impairments to the idler quality during wavelength conversion. Figure 3b presents the bit-error rate (BER) of the back-to-back (B2B) 1,441 nm signal (blue solid line), 1,680 nm signal (yellow solid line) and 1,441 nm idler (purple dotted line). The B2B BER difference between 1,441 and 1,680 nm signals was due to the wavelength-dependent responsivity of the intensity receiver. As can be seen in Fig. 3b, the idler suffered negligible penalty compared to the 1,441 nm B2B signal, which indicates that the proposed CMOS-compatible single-mode nonlinear Si₃N₄ integrated waveguide is promising for all-optical signal processing. In addition, we implemented the all-optical wavelength conversion of single-polarization 32-GBd 16-quadrature-amplitude-modulation (16-QAM) signals with a net rate over 100 Gbit s⁻¹, based on the 56-cm-long single-mode nonlinear Si₃N₄ integrated waveguide. Figure 3c,d show the constellation

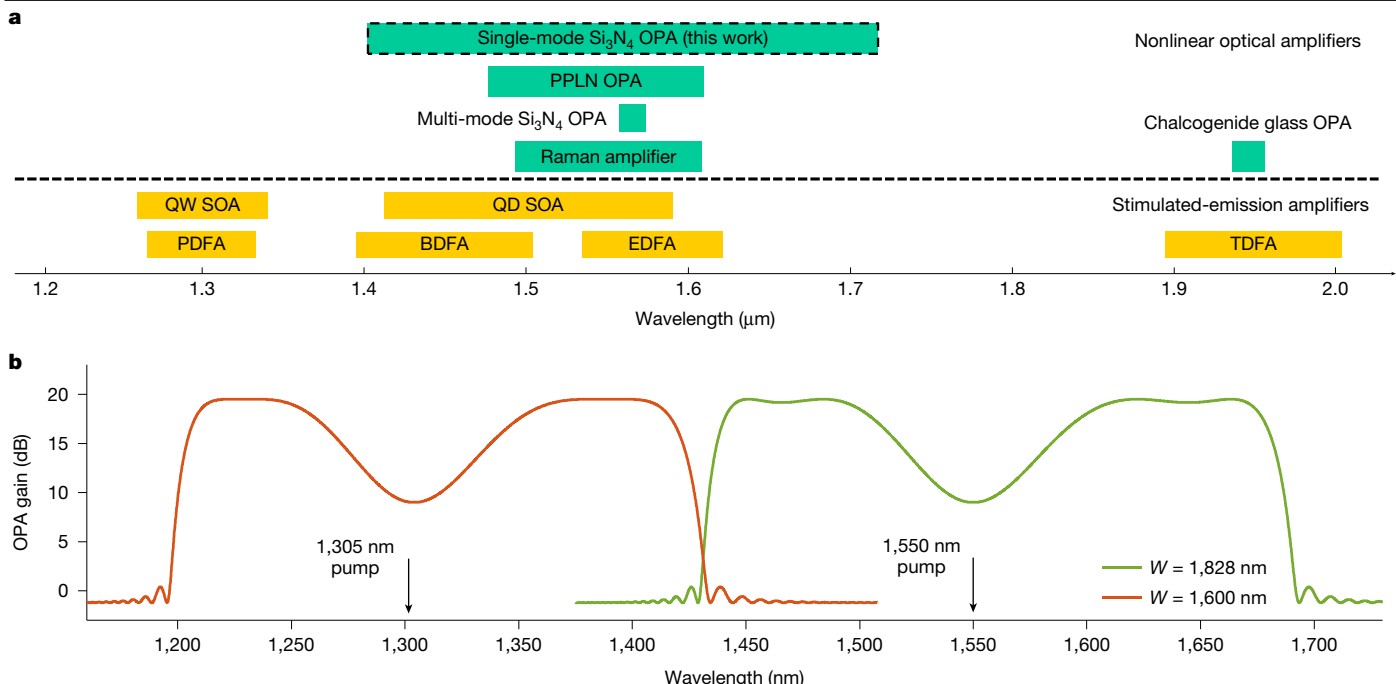

**Fig. 4 | Bandwidth and wavelength ranges of different optical amplifiers.** **a**, State-of-the-art bandwidth of continuous-wave optical amplification in the near-infrared regime based on both stimulated-emission and nonlinear optical platforms. **b**, Theoretical gain spectra of OPAs based on optimized single-mode nonlinear rib $Si_3N_4$ integrated waveguides pumped at wavelengths of 1,305 nm (red) or 1,550 nm (green). Both $Si_3N_4$ waveguides have the same rib and slab thicknesses ($H_1 = 300$ nm and $H_2 = 500$ nm) and can be integrated on the same wafer but for different widths of 1,600 nm and 1,828 nm. The waveguide lengths are 2 m with an assumed propagation loss of 0.6 dB m$^{-1}$. BDFA, bismuth-doped fibre amplifier; EDFA, erbium-doped fibre amplifier; PDFA, praseodymium-doped fibre amplifier; PPLN, periodically poled lithium niobate; QD, quantum dot; QW, quantum well; SOA, semiconductor optical amplifier; TDFA, thulium-doped fibre amplifier.

diagrams for the B2B 1,670-nm signal and converted 1,448-nm idler, respectively. We have realized all-optical wavelength conversion for a more than 200-nm-wide wavelength span at 100 Gbit s$^{-1}$ without amplifying the signal and idler waves. As the 32-GBd 16-QAM is the dominant modulation format of current optical-fibre communication systems connecting the continents on Earth, the $Si_3N_4$-chip high-efficiency wavelength conversion demonstrated has a bright future in the all-optical reconfiguration of global WDM optical networks by unlocking transmission beyond the C and L bands of optical fibres[34] and increasing the capacity of optical neuromorphic computing for artificial intelligence[35].

Figure 4a summarizes the bandwidth of various types of wideband continuous-wave optical amplifiers. Various material platforms have been developed for stimulated-emission optical amplifiers, such as erbium-doped fibre amplifiers[36], bismuth-doped fibre amplifiers[37], thulium-doped fibre amplifiers for optical applications[38], praseodymium-doped fibre amplifiers[39], and quantum-well and quantum-dot semiconductor optical amplifiers[40,41]. To cover the entire transmission windows of telecommunication-grade silica and widely studied hollow-core fibres[42], new stimulated-emission materials with different bandgaps need to be investigated and developed. On the other hand, nonlinear optical effects, including the Raman effect[43], difference-frequency generation and FWM, have also been applied to build wideband optical amplifiers based on single material such that the operating wavelength can be flexibly tuned by changing the pump frequency and the waveguide dispersion. Amplifiers based on the difference-frequency or FWM effects are also called optical parametric amplifiers (OPAs). Periodically poled lithium niobate waveguide OPAs based on the difference-frequency effect have been investigated[44], leading to ultra-high-speed optical-fibre transmission beyond conventional telecommunication bands[45]. Based on the FWM process, OPAs using $\chi^{(3)}$-nonlinear nanophotonic waveguides are free of complicated periodic poling. Continuous-wave nanophotonic OPAs were first realized

with chalcogenide glass microfibres with a gain bandwidth of about 20 nm (ref. 46). Our single-mode nonlinear rib $\chi^{(3)}$ integrated waveguides, which have more freedom in dispersion engineering, enabled us to demonstrate an OPA bandwidth of 330 nm (Fig. 4a), which is wider than that of previously published continuous-wave optical amplifiers. At present, the length of the fabricated 1.9-µm-wide single-mode nonlinear spiral-rib $Si_3N_4$ integrated waveguide greatly limits the parametric gain. We are continuing to optimize the fabrication and believe that the yield of long ultra-low-loss single-mode nonlinear rib $Si_3N_4$ waveguides can be improved. The length issue together with the yield may not be problematic for CMOS foundries, as their deep-ultraviolet exposure technologies are mature for massive nanofabrication.

We theoretically optimized the width of the single-mode nonlinear rib $Si_3N_4$ integrated waveguide for high-gain OPAs. Figure 4b shows the theoretical parametric gain spectra of 2-m-long 0.6 dB m$^{-1}$-loss single-mode nonlinear rib $Si_3N_4$ integrated waveguides with widths of 1,600 nm (red) and 1,828 nm (green) and a pump power of 34 dBm. Both waveguides have the same rib thickness ($H_1 = 300$ nm) and slab thickness ($H_2 = 500$ nm) and can be integrated on the same wafer, potentially with a total chip size of 3 cm × 3 cm. As can be seen in Fig. 4b, the two OPAs, for which the pump wavelengths are 1,305 and 1,550 nm, provide a maximum gain of about 20 dB and cover the whole transmission window of single-mode telecommunication fibres. We also analysed the fabrication tolerance of the high-gain $Si_3N_4$ waveguide OPA (Supplementary Information). With the advances in semiconductor optoelectronics, continuous-wave pump lasers operating at wavelengths of 1,305 and 1,550 nm with watt-level power could also become available[47] and may lead to compact OPAs based on hybrid photonic integration[48]. The spectral flatness of the OPA over the 200-nm bandwidth could be improved by dual-pump, dispersion or pump-phase shifting techniques that can be implemented in integrated waveguide platforms[49,50]. A rib structure with several layers would provide more degrees of freedom

for hyper-dispersion engineering and may lead to a broader parametric bandwidth, which is yet to be explored.

In this paper, we proposed and demonstrated extremely low-loss single-mode nonlinear $Si_3N_4$ integrated waveguides with hyper-dispersion engineering for ultra-broadband, efficient, continuous-wave FWM. Unlike conventional approaches focusing on the waveguide cross section, we exploited the three-dimensional waveguide geometry for on-chip optical field manipulation, simultaneously achieving single-mode transmission and dispersion engineering of nonlinear integrated nanophotonic waveguides in telecommunication bands. Using the 0.56-m-long single-mode nonlinear rib $Si_3N_4$ integrated waveguide with hyper-dispersion engineering, we obtained a continuous-wave gain bandwidth of 330 nm in the near-infrared regime. The whole transmission window of telecommunication silica fibres could potentially be covered by the parametric gain profiles of single-mode nonlinear rib $Si_3N_4$ waveguides integrated on the same wafer. Furthermore, we realized wide all-optical wavelength conversion of single-wavelength signals beyond 100 Gbit $s^{-1}$ without amplifying the signal and idler wave. These experimental results agree with theoretical expectations. A higher continuous-wave parametric gain and conversion efficiency could be expected with longer low-loss single-mode nonlinear rib $Si_3N_4$ integrated waveguides. With the combination of cross-sectional shaping and longitudinal bending, the waveguide design technique we proposed is easy to implement. It could allow other integrated platforms to realize low-loss single-mode dispersion-engineered nonlinear waveguides that could become key building blocks for optics, making the technique promising, from fundamental research in photonics, physics, quantum physics, chemistry and biology to industrial applications in communications, computing, spectroscopy, imaging and metrology.

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

# Article

## Methods

### Fabrication and linear characterization of the spiral-rib waveguides

The proposed rib waveguide was manufactured by the subtractive electron-beam lithography process used for ultra-low-loss high-confinement $Si_3N_4$ waveguides[51,52]. An 800-nm-thick $Si_3N_4$ layer was deposited by low-pressure chemical vapour deposition on a 4-inch Si wafer with 3-μm-thick $SiO_2$ layer on top. Two-step etching was used to fabricate the rib waveguides with dual-layer tapers at the chip edges for coupling with lensed fibres. The 300-nm-thick 1.9-μm-wide $Si_3N_4$ spiral rib was defined in the first etching. To prevent the $Si_3N_4$ from cracking, 3-μm-wide microgrooves between the rib waveguides were formed in the 500-nm-thick slab layer during the second etching. The etched waveguides were then annealed above 1,100 °C in an Ar-flow atmosphere and cladded with 3-μm-thick $SiO_2$ by low-pressure chemical vapour deposition. Finally, the whole wafer was diced into chips by etching. The parameters of WG1 and WG2 are listed in Extended Data Table 1. WG1 and WG2 both consist of several concatenated single-mode spiral units with 800-μm-long straight-connection rib waveguides in between. We used the adiabatic transition between the spiral and connection waveguides. Although the very short straight-connection rib waveguides supported two transverse-electric modes, more than 90% of both WG1 and WG2 operated in single mode. There are 12 and 68 spiral units in WG1 and WG2, respectively.

Lensed fibres with beam-spot diameters of 3 μm were used to couple light with the $Si_3N_4$ nanophotonic chip. We found that the average coupling loss was about 2.5 dB per facet at a wavelength of 1,550 nm for the $TE_{00}$ mode of the spiral-rib $Si_3N_4$ waveguide. The waveguide propagation loss was measured using a commercial OFDR tester combined with a wavelength-scanning laser. For the linear transmission spectrum measurements of the $Si_3N_4$ waveguides, we focused on the $TE_{00}$ mode and used a power-constant tunable laser with a wavelength step of 1 pm.

### FWM characterization with single-mode nonlinear rib $Si_3N_4$ integrated waveguides

We used the pump–probe approach to measure the parametric gain and conversion efficiency of the continuous-wave FWM in the 0.56-m-long single-mode rib nonlinear $Si_3N_4$ integrated waveguide. Three semiconductor external-cavity lasers (ECLs) were used to generate a signal wave that could be tuned from 1,355 to 1,680 nm. Another semiconductor ECL emitted a 1,551.1-nm pump wave, which was amplified by a high-power erbium-doped fibre amplifier. The pump and signal waves were combined by a low-loss thin-film WDM coupler with a bandwidth of 4 nm and entered the single-mode nonlinear rib $Si_3N_4$ integrated waveguide through the lensed fibre. The polarization states of both the pump and signal waves were aligned to the $TE_{00}$ mode of the rib $Si_3N_4$ waveguide. Then, 1% of the optical field at the $Si_3N_4$ waveguide input port was recorded by an optical spectrum analyser. At the $Si_3N_4$ waveguide output port, we used a 15-nm-wide coarse WDM coupler to mitigate the residual pump intensity before we measured the optical spectrum. Using power calibrations for the input and output optical spectra, we calculated the FWM gain and conversion efficiency for each signal wavelength.

### Wavelength conversion for optical communications

Using the 0.56-m-long single-mode nonlinear rib $Si_3N_4$ integrated waveguide, we implemented all-optical wavelength conversion of NRZ and 16-QAM signals. The pump wavelength was 1,551.1 nm with a continuous-wave on-chip power of 34 dBm in all the measurements. No optical amplification was used in the signal or idler wave paths. The green and blue lines in Fig. 3a are for polarization-maintaining and non-polarization-maintaining single-mode fibre patch cords, respectively.

For the NRZ modulation format, a Mach–Zehnder modulator was used to convert the 10 Gbit s$^{-1}$ electrical signals from a BER tester into the optical domain with a carrier wavelength of 1,680 nm. The optical power of the output signal from the Mach–Zehnder modulator was about 2 dBm. After a 10 Gbit s$^{-1}$ NRZ idler wave at 1,441 nm was generated during the on-chip FWM process, two coarse 1,550-nm WDM couplers with bandwidths of 15 nm were used to thoroughly mitigate the residual pump. We used a band-pass filter to select the idler wave. A 10-GHz intensity receiver with one photodetector and two radio-frequency amplifiers converted the optical signals back to the electrical domain, and these signals were fed to the BER tester and used to calculate the BER and record the eye diagrams of the received signal.

Regarding the 16-QAM optical signals, an electrical arbitrary waveform generator was used to generate the 32-GBd in-phase and quadrature components, which were amplified separately and sent to a single-polarization coherent electrical-optical modulator. To detect the 16-QAM optical signals after wavelength conversion, a commercial coherent receiver with another tunable ECL as a local oscillator was used. The signal wavelength at the chip input was set to 1,670 nm so that the 16-QAM data could be converted to a 1,447-nm idler wave such that the wavelength-dependent coherent receiver responsivity was still sufficient for data recovery. The electrical signals after the coherent receiver were recorded by a high-speed real-time scope. Offline digital signal processing was applied to analyse the signal BER and constellation.

## Data availability

The data that support the findings of this study are available from the corresponding authors upon request.

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

**Acknowledgements** We thank M. Karlsson and Y. Gao for fruitful technical discussions, E. Torres, M. Myremark and C. Lei for their help with chip preparation and experiments, M. Pu at the Technical University of Denmark, F. Olofsson, and the Santec Corporation for sharing equipment. This work was performed in part at Myfab Chalmers. This work was partially funded by the Swedish Research Council (grant nos. VR-2015-00535 to P.A.A. and VR-2020-00453 to V.T.-C.), the K. A. Wallenberg Foundation through a KAW Scholarship (P.A.A.) and the Chinese Fundamental Research Funds for the Central Universities (grant no. YJ202478).

**Author contributions** P.Z. and P.A.A. were responsible for the concept. P.Z. designed the waveguide, characterized the chip, performed the theoretical modelling of the waveguides, simulated the nonlinear process, designed the waveguide layout, completed the linear and nonlinear characterization of the waveguides, and processed the data. V.S. and M.G. developed the waveguide nanofabrication process. P.Z., Z.H. and V.S. used the chip for optical transmission and implemented the wavelength conversion experiment for optical communications. V.S. fabricated the waveguides. P.A.A. and V.T.-C. acquired funding and administered the project. P.A.A. supervised the project. P.Z. and P.A.A. wrote the original draft of the manuscript. P.A.A., V.S. and V.T.-C. reviewed and edited the manuscript.

**Funding** Open access funding provided by Chalmers University of Technology.

**Competing interests** The authors declare no competing interests.

**Additional information**
**Correspondence and requests for materials** should be addressed to Ping Zhao or Peter A. Andrekson.

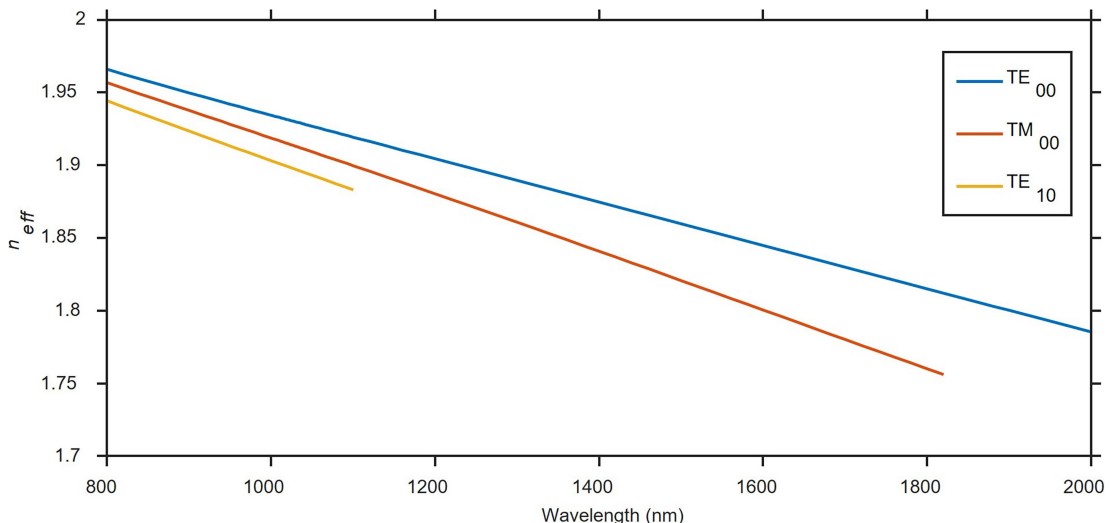

**Extended Data Fig. 1 | Simulated effective refractive index varying with wavelength for a bent single-mode rib silicon nitride waveguide.** The waveguide bending radius in the simulation based on COMSOL Multiphysics is constant 400 μm with W = 1.9 μm, $H_1$ = 300 nm and $H_2$ = 500 nm. As can be seen, $TE_{10}$ and $TM_{00}$ modes are cut off at the wavelength of 1100 nm and 1820 nm, respectively. Thus, only one TE and one TM mode exists at the wavelength range studied here.

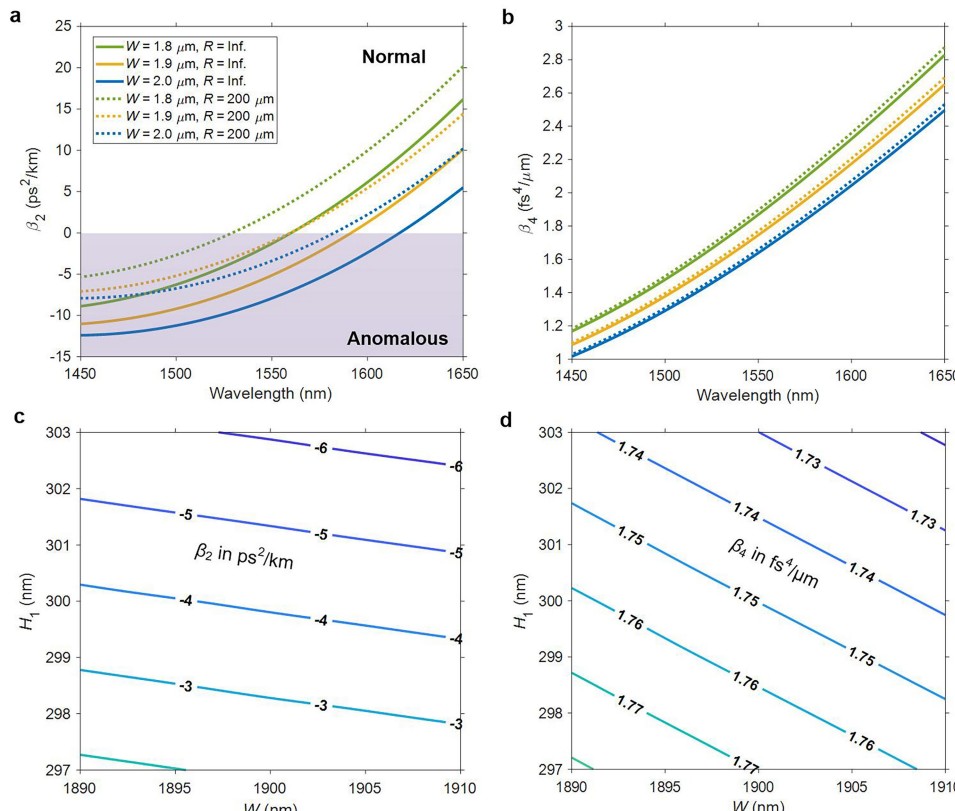

**Extended Data Fig. 2 | Dispersion properties of rib Si₃N₄ nonlinear integrated waveguides. a,b**, Second- (**a**) and fourth-order (**b**) dispersion spectra of TE₀₀ mode in the proposed rib Si₃N₄ nonlinear integrated waveguides with H₁ = 300 nm and H₂ = 500 nm. The solid and dotted lines correspond to cases of straight waveguides and bent waveguides with a radius of 200 μm, respectively.

The green, yellow and blue lines are for the cases of W = 1.8 μm, 1.9 μm and 2.0 μm, respectively. **c,d** Second- (**c**) and fourth-order (**d**) dispersion at 1550 nm wavelength varying with W and H₁ when the radius of a single-mode rib Si₃N₄ nonlinear integrated waveguide is 400 μm for fabrication tolerance analysis.

**Extended Data Table 1 | Parameters of spiral rib Si$_3$N$_4$ nonlinear integrated waveguides fabricated**

| Waveguide | Length (cm) | Rib gap (µm) | Slab width (µm) | Minimum spiral bending radius (µm) | Maximum spiral bending radius (µm) | Length of spiral unit (mm) | Chip size (mm$^2$) |
|---|---|---|---|---|---|---|---|
| WG 1 | 18 | 15 | 12 | 180 | 450 | 14.2 | 1 x 14 |
| WG 2 | 56 | 50 | 30 | 165 | 450 | 7.4 | 3 x 29 |