## [Peer Review File · Nature]

Ultra-broadband optical amplification using nonlinear integrated waveguides

Corresponding Author: Professor Peter Andrekson

Version 0:

Reviewer comments:

Referee #1

(Remarks to the Author)

This article presents the demonstration of an integrated optical parametric amplifier with an ultra-wide bandwidth of 330 nm and an on/off gain exceeding 2 dB. This performance was achieved using a silicon nitride integrated waveguide with an unprecedentedly low loss of 2.5 dB/m. The authors designed the waveguide in a spiral configuration and carefully tuned the parameters to exploit bending loss, enabling single-mode operation. Without this optimization, the linear losses would be too high to achieve such a broad bandwidth. By further optimizing the β_2/β_4 ratio, they report ultra-large parametric gain bands. Using a high-power continuous-wave (CW) laser, they demonstrate the amplifier's performance in terms of gain and conversion efficiency for both CW seed signals and high-bit-rate telecommunications signals. The results show excellent amplification quality, making these amplifiers compliant with telecommunications standards. Their experimental findings align well with numerical and analytical models, indicating strong control over system parameters. In the final section, they project that a gain of 20 dB could be achieved with longer waveguides, aligning with telecommunications requirements.

The article is well-written and illustrated, clearly detailing the significance of this work and effectively positioning it within the state of the art. Each step—design, fabrication, and parametric processes—is thoroughly mastered and described, lending credibility to the authors' projection of producing longer waveguides. The performance is impressive, particularly in terms of bandwidth, far surpassing that of rare-earth-doped systems. Although the current gain values are moderate, this work demonstrates the promising potential of integrated parametric amplifiers for future telecommunications applications.

I recommend publishing the paper despite the current moderate gain value. The article is well-written and clear. I have only minor comments for consideration prior to publication:

1. Have the authors experimentally verified that the waveguide operates in single mode? Was the cutoff frequency measured?
2. While the authors select various parameters to describe the spiral, they do not account for the spacing between waveguides. Could this parameter also be considered?
3. In line 138, is the specified pump power of 35 dBm measured outside the waveguide?
4. What are the saturation properties of the amplifier? Does this introduce significant differences compared to fiber-based systems?
5. Would it be possible to achieve larger bandwidths by using other structures to further reduce the β_4 value?
6. While loss reduces the pump conversion efficiency, it could potentially help to flatten the gain band. Could this approach be considered?

Referee #2

(Remarks to the Author)

The work is clearly presented and the manuscript is readable. The idea that waveguide bends strip higher-order modes is known from the 1960s (Marcatili, Marcuse etc.) and is not controversial. There is some terminology confusion as this is not truly a single mode waveguide (as shown by the electromagnetic density of modes) but merely measures as one because

the other modes have higher loss and the waveguide is long. Pump power is still lost to those modes. Low loss Si₃N₄ waveguides have been around for some years and are commercially available with similar loss numbers (few dB/m) from several places. The communications experiments are acceptable. With high pump power (Watts), the on-off gain is above unity. The combination of this device concept and this material might not be capable of much more and so this paper represents a technical accomplishment. This paper has good results which are clearly presented and could be published in journals like the IEEE Journal of Lightwave Technology, Optics Express or Nature.

Referee #3

(Remarks to the Author)

The authors report on ultra-broad wavelength conversion using four-wave-mixing in silicon nitride rib waveguides. The work is technically solid, well written and well presented. The experimental results are very impressive both in terms of conversion efficiency and the bandwidth over which it is achieved. There are however some concerns that should be addressed before further decisions can be made.

1) In the abstract the authors say that “conventional approaches of nonlinear photonic waveguide geometry construction for dispersion engineering focus on waveguide cross section and result in always being multimode as a byproduct.” There have been other works on four-wave-mixing that use rib mode waveguides, which are apparently single-mode. See for example, <https://doi.org/10.1109/LPT.2013.2272521>, <https://doi.org/10.1364/OE.17.003514>, <https://doi.org/10.1364/OL.41.002418>
The authors should clearly explain what the conceptual novelty of their work is compared to previous rib-based approaches. On the other hand, curving waveguides to eliminate higher order modes is a very well-known approach.

2) Fig. 1(c) would be more illustrative if the authors could show the curvature losses associated with each mode in a logarithmic scale. This would enable the reader to see that the TE₁₀ mode is effectively cut-off without (significantly) affecting the losses of the TE₀₀ mode.

3) In Fig. 1(e) it is hard to see the small losses, i.e. the slopes of the traces, due to the large dynamic range of the measurement. It would be preferable to show a more limited range and clearly identify the slopes by showing straight lines fitted to the data.

4) Fabrication tolerances are given in Fig. E1(c) and (d) in terms of β_2 and β_4 , but especially the values of β_4 shown in Fig. E1 are quite different from the optimum values of Fig. 2(a). It would be very insightful to analyze via simulations how fabrication tolerances affect the gain bandwidth and the conversion bandwidth (Figs. 2(c) and (d)). In other words, how tightly have waveguide dimensions to be controlled to achieve the desired hyper-dispersion engineering and, hence, the reported results?

5) The discussion is a bit lengthy and could be condensed. The OPA gain results in the discussion are simulations, but are not accompanied by experimental validation. It would be desirable to assess the fabrication tolerances required to achieve these results (like in the previous point).

6) In line 69, Fig. 1(a) seems to refer to Fig. 1(c).

7) In line 144, Fig. 1(e) seems to refer to Fig. 2(a).

Version 1:

Reviewer comments:

Referee #1

(Remarks to the Author)

The authors have responded satisfactorily to my comments. I recommend publication of the document in its present form.

Referee #2

(Remarks to the Author)

I have reviewed and considered the authors' responses, and do not have further comments or questions, reiterating my earlier observations and (positive) assessment. To summarize:

- Even if the major concepts of bent waveguides stripping high-order modes are known, there is some achievement in this work to actually fabricate the bends.

- Even if a high level of mixer gain has not been shown, at least some positive gain has been achieved, based on the ideas (known especially from fiber parametric optics) that β_2 and β_4 are both important along with low loss and high power.

- Low loss Si₃N₄ waveguides have been around for some years now including commercial sources with similar loss numbers (few dB/m), so the achievements here are easily believable. The communications experiments are also acceptable.

Overall, this paper has good results of progress in this topic using solid scientific principles which are clearly presented and could be published in any journal.

Referee #3

(Remarks to the Author)

This is a re-review of a manuscript that I already reviewed.

In the revised version the authors have addressed the concerns I raised in my original review. Especially the tolerance analysis they provide is very insightful and highlights the challenges associated with achieving the desired hyper-dispersion engineering.

We thank the reviewers for the very useful feedback on our manuscript. Here we provide a detailed response in red, describe the actions taken in the manuscript and marked the revision in the manuscript in yellow. We here did not mark minor changes for clarity.

Thanks also for the feedback from the Senior Editor. We have rewritten the introductory paragraph accordingly.

Referee #1 (Remarks to the Author):

This article presents the demonstration of an integrated optical parametric amplifier with an ultra-wide bandwidth of 330 nm and an on/off gain exceeding 2 dB. This performance was achieved using a silicon nitride integrated waveguide with an unprecedentedly low loss of 2.5 dB/m. The authors designed the waveguide in a spiral configuration and carefully tuned the parameters to exploit bending loss, enabling single-mode operation. Without this optimization, the linear losses would be too high to achieve such a broad bandwidth. By further optimizing the β_2/β_4 ratio, they report ultra-large parametric gain bands. Using a high-power continuous-wave (CW) laser, they demonstrate the amplifier's performance in terms of gain and conversion efficiency for both CW seed signals and high-bit-rate telecommunications signals. The results show excellent amplification quality, making these amplifiers compliant with telecommunications standards. Their experimental findings align well with numerical and analytical models, indicating strong control over system parameters. In the final section, they project that a gain of 20 dB could be achieved with longer waveguides, aligning with telecommunications requirements.

The article is well-written and illustrated, clearly detailing the significance of this work and effectively positioning it within the state of the art. Each step—design, fabrication, and parametric processes—is thoroughly mastered and described, lending credibility to the authors' projection of producing longer waveguides. The performance is impressive, particularly in terms of bandwidth, far surpassing that of rare-earth-doped systems. Although the current gain values are moderate, this work demonstrates the promising potential of integrated parametric amplifiers for future telecommunications applications.

I recommend publishing the paper despite the current moderate gain value. The article is well-written and clear. I have only minor comments for consideration prior to publication:

1. Have the authors experimentally verified that the waveguide operates in single mode? Was the cutoff frequency measured?

We verified the single mode property of the proposed waveguides through the following three aspects:

1) There were negligible ripples on the transmission spectrum of the waveguide we fabricated (Fig 1a, red data). In contrast, strong spectral ripples due to modal coupling in TE polarization were observed in few-mode waveguides with the same length as can be seen in Fig. 1 (a) in our manuscript, blue data. Similar phenomena were reported in Ref. 37 as well.

2) The theoretical and experimental pump powers matched well in the four-wave mixing measurements for the proposed waveguide, while there was a 3 dB difference for few-mode silicon nitride nonlinear integrated waveguides in our previous work (Ref. 36).

3) The polarization extinction ratio of optical signals at the rib waveguide output was more than 20 dB.

We have, however, not measured the cutoff frequency. We determined the cutoff frequency/wavelength via simulations based on COMSOL Multiphysics. The figure below shows the simulated effective refractive index as a function of wavelength for a waveguide bending radius of $400\ \mu\text{m}$ with $W = 1.9\ \mu\text{m}$, $H_1 = 300\ \text{nm}$ and $H_2 = 500\ \text{nm}$. As can be seen, TE_{10} and TM_{00} modes are cut off at the wavelength of $1100\ \text{nm}$ and $1820\ \text{nm}$, respectively. Thus, only one TE and one TM mode exists at the wavelength range studied here.

Modification in the manuscript: We added an extended figure, i.e., Fig. E1, in the main text as below.

Figure E1. Simulated effective refractive index varying with wavelength for a bent single-mode rib silicon nitride waveguide. The waveguide bending radius in the simulation based on COMSOL Multiphysics is constant $400\ \mu\text{m}$ with $W = 1.9\ \mu\text{m}$, $H_1 = 300\ \text{nm}$ and $H_2 = 500\ \text{nm}$. As can be seen, TE_{10} and TM_{00} modes are cut off at the wavelength of $1100\ \text{nm}$ and $1820\ \text{nm}$, respectively. Thus, only one TE and one TM mode exists at the wavelength range studied here.

2. While the authors select various parameters to describe the spiral, they do not account for the spacing between waveguides. Could this parameter also be considered?

Yes, this parameter is important and has been considered in our manuscript. The coupling between adjacent waveguides should be avoided. We describe in the Supplementary how the slab width of the waveguide affects the effective refractive index of TE_{00} mode and used a slab width large enough to make TE_{00} mode undisturbed by other sections of the spiral.

Modification in the Supplementary: We added a clarification in the supplementary and marked it in yellow, Line 46-48: “Besides, the spacing between adjacent rib spiral waveguides equals the slab width plus the width of the groove in the slab and is sufficiently large so that coupling in each spiral unit is avoided.”

3. In line 138, is the specified pump power of 35 dBm measured outside the waveguide?

This refers to Figure 2a (theoretical analysis). The pump power that enters the waveguide is 35 dBm, and this is now clarified in the manuscript.

Modification in the Manuscript, Line 129-130: "Fig. 2(a) depicts the theoretical CE spectra of a 1-m-long nonlinear integrated waveguide with a pump power of 35 dBm at 1550 nm."

4. What are the saturation properties of the amplifier? Does this introduce significant differences compared to fiber-based systems?

Since the gain was moderate and the signal input power was small, we did not observe any saturation of the amplifier. If they are operated in a saturation condition, our waveguide amplifiers will behave similarly to the case of nonlinear fibers which has been studied previously.

5. Would it be possible to achieve larger bandwidths by using other structures to further reduce the β_4 value?

We studied different rib waveguide dimensions to maximize the gain bandwidth. We changed the parameters of the rib waveguide and obtained a reduced β_4 value. However, β_2 increased and limited further bandwidth expansion. Alternative configurations, such as multi-layer silicon nitride rib structures, may lead to further-improved hyper-dispersion engineering, but they are yet to be explored.

Modification in the Manuscript, line 226-228: "Multiple layers with the rib structure provide more degrees of freedom for hyper dispersion engineering and may lead to broader parametric bandwidth, which is yet to be explored. "

6. While loss reduces the pump conversion efficiency, it could potentially help to flatten the gain band. Could this approach be considered?

Yes, it would be useful. We investigated this loss effect on silicon nitride waveguide parametric amplifiers (Optics Express 28(16), pp. 23467-23477 (2020)). The phase matching condition changes due to the loss of pump power. In this manner, the gain bandwidth may be flattened.

Modification in the Manuscript: we cited one additional paper [68] in the discussion section, Line 226: "...dual-pump, dispersion or pump-phase shifting techniques which can be implemented in integrated waveguide platforms [66-68]."

Referee #2 (Remarks to the Author):

The work is clearly presented and the manuscript is readable. The idea that waveguide bends strip higher-order modes is known from the 1960s (Marcatili, Marcuse etc.) and is not controversial. There is some terminology confusion as this is not truly a single mode waveguide (as shown by the electromagnetic density of modes) but merely measures as one because the other modes have higher loss and the waveguide is long. Pump power is still lost to those

modes. Low loss Si₃N₄ waveguides have been around for some years and are commercially available with similar loss numbers (few dB/m) from several places. The communications experiments are acceptable. With high pump power (Watts), the on-off gain is above unity. The combination of this device concept and this material might not be capable of much more and so this paper represents a technical accomplishment. This paper has good results which are clearly presented and could be published in journals like the IEEE Journal of Lightwave Technology, Optics Express or Nature.

We agree that the idea of using bends to achieve a single-mode waveguide condition is well known. We now elaborate on this in the revised main text and cite appropriate papers and books.

To the best of our knowledge, all previous integrated $\chi^{(3)}$ -driven parametric amplification demonstrations were based on **multimode** nonlinear integrated waveguides with **anomalous dispersion**. Anomalous dispersion is a key factor to achieve high parametric gain and ultrabroad bandwidth, while normal dispersion results in narrow bandwidth as well as low parametric gain, as can be seen in the updated Supplementary (When we address the concern from Reviewer #3 on the fabrication tolerance analysis, we also made a comparison of parametric gain spectra between normal and anomalous dispersion). The mode coupling in multimode nonlinear waveguides results not only in the reduction FWM efficiency and bandwidth but also spectral ripple in the transfer function causing distortion of modulated signals. Single-mode nonlinear integrated waveguides featuring anomalous dispersion as presented here for the first time can solve these fundamental issues.

We kept most of the whole waveguide in a single-mode condition using the bending. To obtain meter-long silicon nitride nonlinear integrated waveguides, we used very short straight connection rib waveguides to connect multiple spirals of which the maximum radii were less than 450 μm and within the single-mode regime, i.e., yellow shaded area in Fig. 1(d). The connection rib waveguides between adjacent spiral units were 800 μm long and supported two modes in TE polarization, while the waveguide length in each spiral unit was about 7.4 mm for WG 2. Hence, 90% of the 56-cm-long rib silicon nitride waveguide with 68 spiral concatenated units was single mode in each polarization. The transition between the spiral and connection waveguides is adiabatic.

The measured polarization extinction ratio of broadband optical signals at the 0.56-meter-long rib waveguide output was more than 20 dB, indicating negligible coupling between TE and TM modes. In addition, the waveguide FWM characterization agrees well with the theoretical expectations using the same pump power, which indicates that most pump power remains in the TE₀₀ mode along the single-mode rib waveguide, while the pump power difference between theory and measurement was up to 3 dB for few-mode silicon nitride waveguides (Ref. 36).

The amplification is modest and limited by the waveguide length. With 2 meter long such low-loss nonlinear waveguides, it would lead to a gain of 20 dB, as shown by Fig. 4(b).

Modification in the Manuscript:

We highlighted the novelty of our work in the revised introductory paragraph and main text:

Line 65-67

“Anomalous dispersion ($\beta_2 < 0$) is of vital importance to realize phase matching ($\Delta K = 0$) for high parametric gain and wideband operation, as can be seen in Supplementary.”

Line 94-96

“The key technique here to simultaneously achieve single-mode operation and anomalous dispersion is bending the waveguide to cut off higher-order modes and maintain the anomalous dispersion.”

We cite these papers and books. Line 92

“Rib waveguides with silica cladding are used to achieve fewer guiding modes as well as lower propagation losses, compared to rectangular-core waveguides with the same width and total thickness [44-46].”

We also added more details of the waveguide in the Methods: Line 261-266

“WG 1 and 2 both consist of concatenated single-mode spiral waveguides with 800- μm -long straight connection rib waveguides in between. We used adiabatic transition between the spiral and connection waveguides. Although the very short straight connection rib waveguides supported two TE modes, more than 90 percent of both WG 1 and WG 2 is single mode. There are 12 and 68 spiral units in WG1 and WG2, respectively.”

Table 1. Parameters of proposed spiral rib Si_3N_4 nonlinear integrated waveguides

Waveguide	Length (cm)	Rib gap (μm)	Slab width (μm)	Minimum spiral bending radius (μm)	Maximum spiral bending radius (μm)	Length of spiral unit (mm)	Chip size (mm^2)
WG 1	18	15	12	180	450	14.2	1 x 14
WG 2	56	50	30	165	450	7.4	3 x 29

Referee #3 (Remarks to the Author):

The authors report on ultra-broad wavelength conversion using four-wave-mixing in silicon nitride rib waveguides. The work is technically solid, well written and well presented. The experimental results are very impressive both in terms of conversion efficiency and the bandwidth over which it is achieved. There are however some concerns that should be addressed before further decisions can be made.

1) In the abstract the authors say that “conventional approaches of nonlinear photonic waveguide geometry construction for dispersion engineering focus on waveguide cross section and result in always being multimode as a byproduct.” There have been other works on four-wave-mixing that use rib mode waveguides, which are apparently single-mode. See for example, <https://doi.org/10.1109/LPT.2013.2272521>, <https://doi.org/10.1364/OE.17.003514>, <https://doi.org/10.1364/OL.41.002418>

The authors should clearly explain what the conceptual novelty of their work is compared to previous rib-based approaches. On the other hand, curving waveguides to eliminate higher order modes is a very well-known approach.

Thanks for sharing these relevant papers. The novelty of our work is a waveguide that simultaneously leads to an effective single-mode behavior, anomalous dispersion and ultralow-loss. The combination of these three elements is essential to attain ultrabroadband parametric gain and to the best of our knowledge it has not been attained before (see reply to reviewer #2). The approach of designing single-mode dispersion-engineered nonlinear waveguides in this paper is universal, in that it may be applied to many kinds of waveguides.

The waveguides in both Paper One (<https://doi.org/10.1109/LPT.2013.2272521>) and Three (<https://doi.org/10.1364/OL.41.002418>) exhibited very large normal dispersion in the single mode regime and resulted in FWM bandwidths less than 20 nm. We already cited one paper (Ref. 16) from the group who used almost the same rib chalcogenide waveguides with two TE modes and published Paper Two (<https://doi.org/10.1364/OE.17.003514>). In addition, due to high nonlinear losses, the integrated chalcogenide waveguides could only be operated with pulsed pumps to achieve parametric gain (Ref. 16).

Modification in the Manuscript: We cited these papers and highlighted the novelty of this work (like the reply to Reviewer 2).

Line 52-54: “Many kinds of such nonlinear platforms have been explored [30], including silicon [11] [31-35], ... , nonlinear glasses [16][17][39],...”

Line 65-67 “Anomalous dispersion ($\beta_2 < 0$) is of vital importance to realize phase matching ($\Delta K = 0$) for high parametric gain and wide band, as can be seen in Supplementary.”

Line 94-96 “The key technique to achieve simultaneous single-mode operation and anomalous dispersion is bending of the waveguide to cut off higher-order modes and maintain the anomalous dispersion.”

2) Fig. 1(c) would be more illustrative if the authors could show the curvature losses associated with each mode in a logarithmic scale. This would enable the reader to see that the TE₁₀ mode is effectively cut-off without (significantly) affecting the losses of the TE₀₀ mode.

Thanks for this good advice. We did simulations for TE₀₀ and TE₁₀ modes transiting from a straight rib waveguide to a single-mode bend rib waveguide, respectively.

Modification in the Manuscript: Added this point in Line 108-109: “We also show how the mode propagation of TE₀₀ and TE₁₀ modes is affected by bending in the **Supplementary**.”

Modification in the Supplementary: We added a section, i.e., Sec. VI, on mode propagation with and without waveguide bending, including the figure below. This simulation illustrated how the TE₁₀ mode is experiencing a large propagation loss.

3) In Fig. 1(e) it is hard to see the small losses, i.e. the slopes of the traces, due to the large dynamic range of the measurement. It would be preferable to show a more limited range and clearly identify the slopes by showing straight lines fitted to the data.

Thanks for this suggestion. We added straight lines fitted to the data to identify the trace slope. Modification in the Manuscript: Added this point in Line 118.

“Indicated by the red slope line in Fig. 1(e), the measured propagation loss of WG 1 is 0.6 dB/m...”

4) Fabrication tolerances are given in Fig. E1(c) and (d) in terms of β_2 and β_4 , but especially the values of β_4 shown in Fig. E1 are quite different from the optimum values of Fig. 2(a). It would be very insightful to analyze via simulations how fabrication tolerances affect the gain bandwidth and the conversion bandwidth (Figs. 2(c) and (d)). In other words, how tightly have waveguide dimensions to be controlled to achieve the desired hyper-dispersion engineering and, hence, the reported results?

This is a good suggestion of analyzing the bandwidth variation with fabrication tolerance. Fig. 2(a) is an illustration of how the hyper-dispersion engineering improves the FWM bandwidth. The dispersion in Fig. 2(a) does not correspond to the waveguide geometry.

The fitted β_2 ($-2.2 \text{ ps}^2/\text{km}$) is slightly smaller than the designed value of $-4 \text{ ps}^2/\text{km}$. β_2 is more sensitive to thickness variation, while β_4 exhibits larger fabrication tolerance, as can be seen from Fig. E1(c) and (d). The thickness of the silicon nitride layer after LPCVD from the center to the edge of the 4-inch wafer changes according to fabrication data from our cleanroom. The 0.56-m-long single-mode silicon nitride waveguide covered a chip area of 3 cm x 3 mm. Consequently, the difference between theoretical and fitted β_2 may be attributed to the thickness change across the waveguide. In the future, we will planarize the SiN wafer using chemical mechanical polishing which is a mature technique and can lead to a surface variation within 0.2 nm (<https://doi.org/10.1364/OPTICA.4.000619>).

We also presented the dispersion variations within a small ranges of rib width ($1900 \text{ nm} \pm 10 \text{ nm}$) and height ($300 \text{ nm} \pm 3 \text{ nm}$) which are shown by extended figures Fig. E2 (c) and (d). This is an analysis on the fabrication tolerance for the dispersion of both β_2 and β_4 .

Modification in the Manuscript: We added the discrepancy analysis of dispersion to the experimental results. Line 163-167: “ β_2 is more sensitive to thickness variation, while β_4 exhibits a larger tolerance to dimension variations, as can be seen from Fig. E2(c) and (d). The fitted β_2 is

slightly smaller than the designed value of $-4 \text{ ps}^2/\text{km}$, which is mainly attributed to the thickness change on the waveguide. Planarization of Si_3N_4 wafers will be utilized to improve the thickness uniformity in the future [50].”

Modification in the Manuscript: We pointed out the fabrication tolerance analysis on dispersion shown by Fig. E2 (c) and (d). Line 112-113: “Fig. E2 (c) and (d) present the second- and forth-order dispersion varying with small rib dimension variations, respectively, i.e., fabrication tolerance analysis.”

5) The discussion is a bit lengthy and could be condensed. The OPA gain results in the discussion are simulations, but are not accompanied by experimental validation. It would be desirable to assess the fabrication tolerances required to achieve these results (like in the previous point).

We added a tolerance analysis of the rib width and height for optimized high parametric gain around 1550 nm wavelength, respectively, which is shown by the following figures in the modifications (also new Fig. S5).

Modification in the Manuscript: We have condensed the discussion section. Line 189-212.

Modification in the Manuscript: We have pointed out the fabrication tolerance analysis. Line 220-221: “We also analyse the fabrication tolerance of the high-gain Si_3N_4 waveguide OPA (Supplementary).”

Modification in the Supplementary: We added this fabrication tolerance analysis, Line 79-91.

“V. Fabrication tolerance to OPA spectra

Figure S5 (a) and (b) show a theoretical analysis of tolerance on the rib width and height for a silicon nitride waveguide OPA, respectively. The optimized gain spectrum is the red solid curve with a rib width of 1828 nm and a rib thickness of 300 nm. The waveguide loss is 0.6 dB/m and the length of the waveguide is 2 m with a pump power of 34 dBm. We consider a typical width uncertainty of ± 5 nm for EBL. When the width increases from 1823 nm to 1833 nm, the corresponding bandwidth as well as the gain flatness changes, as can be seen in Fig. S5 (a). When the rib width is 1780 nm, the second-order dispersion becomes normal ($\beta_2 > 0$). In this case, phase matching is not satisfied and results in a low gain/CE and a narrow bandwidth. The OPA spectrum is more sensitive to the height variation, as can be seen in Fig. S5 (b), which is due to that the waveguide has stronger field confinement in height direction. Wafer planarization techniques, such as chemical mechanical polishing which can lead to a surface roughness down to 0.2 nm [5], would be useful to fabricate wideband high-gain waveguide OPAs.”

Figure S5. Theoretical gain spectra of single-mode silicon nitride rib waveguide OPAs with fabrication uncertainties in waveguide (a) width and (b) height. The slab thickness is 500 nm with a CW 1550 nm pump power of 34 dBm. The waveguide loss is 0.6 dB/m and the length of the waveguide is 2 m.

6) In line 69, Fig. 1(a) seems to refer to Fig. 1(c).

7) In line 144, Fig. 1(e) seems to refer to Fig. 2(a).

Thanks for pointing out these typos which have been corrected.

We also updated the reference list as highlighted in the manuscript.